# Blood Parameters and Feline Tooth Resorption: A Retrospective Case Control Study from a Spanish University Hospital

**DOI:** 10.3390/ani11072125

**Published:** 2021-07-17

**Authors:** Ana Whyte, María Teresa Tejedor, Jaime Whyte, Luis Vicente Monteagudo, Cristina Bonastre

**Affiliations:** 1Department of Animal Pathology, Faculty of Veterinary Medicine, University of Zaragoza, C/Miguel Servet 177, 50013 Zaragoza, Spain; awhyte@unizar.es (A.W.); cbonastr@unizar.es (C.B.); 2Department of Anatomy, Embryology and Animal Genetics, Faculty of Veterinary Medicine, University of Zaragoza, C/Miguel Servet 177, 50013 Zaragoza, Spain; ttejedor@unizar.es; 3CIBER CV (University of Zaragoza—IIS), Faculty of Veterinary Medicine, University of Zaragoza, C/Miguel Servet 177, 50013 Zaragoza, Spain; 4Department of Human Anatomy and Histology, Faculty of Medicine, University of Zaragoza, C/Domingo Miral, s/n, 50009 Zaragoza, Spain; jwhyte@unizar.es

**Keywords:** resorption, feline, blood parameters, case control study

## Abstract

**Simple Summary:**

Tooth resorption (TR) is a progressive destruction of hard dental tissues, leading to dental fractures. Our aims were to describe the TR clinical presentation on data from a university veterinary hospital (September 2018–May 2019; Northeastern Spain), and to study several blood parameters (34) for ascertaining potential systemic effects associated with TR. Cases (29) had positive radiographic TR diagnosis and controls (58) showed healthy mouths when presented for elective surgery; orthopedic surgery or soft tissues procedures. Blood parameters significantly different for cases and controls were chosen for multiple regression analysis (correction factor: age). TR was detected in 130/870 teeth (14.9%). TR stage 4 and 5; and types 1 and 2 were the most frequent. The status of LLP1, LRP1, and LLM1could be considered as TR sentinels. A significant association was found between TR stage and TR type (*p <* 0.001). TR presence was significantly associated with high creatinine levels and low urea nitrogen/creatinine and albumin/globulin ratios. A positive association was found between TR index and globulin levels. When affected by TR, systemic implication related to infection/inflammation or even kidney damage could be present; therefore, special care in these aspects must be provided in feline clinics.

**Abstract:**

Tooth resorption (TR; progressive destruction of hard dental tissues) varies in prevalence according to population, age, and country (29–66.1%). Our objective was twofold: describing the TR clinical presentation in Northeastern Spain, and studying 34 blood parameters to ascertain potential systemic effects associated with TR. Cases (29; presented from September 2018 to May 2019) and controls (58) were considered. Non-parametric tests were carried out to compare cases and controls for each blood parameter; those showing significant differences were chosen for multiple regression analysis (binomial logistic and hierarchical multiple regressions). In case TR was detected in 130/870 teeth (14.9%), TR stage and type were correlated (*p <* 0.001). Increasing CREA values (*p =* 0.034) and decreasing BUN/CREA and ALB/GLOB values were associated with TR presence (*p =* 0.029 and *p =* 0.03, respectively). Increasing GLOB was associated with increasing severity of TR (*p <* 0.01). Type 1 TR (highly related to inflammation and periodontal disease PD) was the most frequently observed type; the association of TR and inflammation biomarkers (ALB/GLOB, GLOB) are explained by this fact. The concomitant presence of PD and TR in old cats would cause TR association with kidney damage biomarkers (CREA, BUN/CREA). When affected by TR, special care in these aspects must be provided to cats.

## 1. Introduction

Tooth resorption (TR) is progressive destruction of the cementum, dentin, and enamel by odontoclastic action, which can result in dental fracture [1]. Studies on cat populations (healthy individuals or patients with oral diseases), age [2], and even geographic region of origin, have shown that tooth resorption (TR) prevalence in cats varies widely. In a clinically healthy population from England, with an average age of 4.9 years, the TR prevalence was 29% [3]. In Minnesota, TR lesions were found in 48% of cats (older than 1 year) submitted to anaesthesia for various procedures [4]. In a retrospective study (1995–1998) carried out in California, 60.8% of feline dental patients showed TR lesions [5]. In a recent study, we found TR lesions in 66.1% of cats treaded at the Odontology service in our University Veterinary hospital (Northeastern Spain) [6].

Inflammatory cytokines can stimulate odontoclastic activity [7,8]. Minor tooth trauma can damage both the periodontal ligament and the cementoblast layer, with the release of osteoclastic activating factors at the site of injury [9]. Also, TR may occur as an idiopathic process [10]. Occasionally, the dental hard tissues are replaced by bone due to osteoblast activity [11].

A series of predisposing factors has been proposed for TR. Increasing age has been associated to TR [3,12]. Bacteria in dental plaque could initiate non-inflammatory TR or even transform it into inflammatory TR [11]. Diet can also play a decisive role in TR occurrence. The high acidity in dry industrial feed harms both enamel and cementum; even acid regurgitation of hairballs could cause lesions in these hard tissues [12,13]. Both the reduced stimulus indoors and the soft consistency of commercial cat food would affect the functional integrity of the periodontal ligament, leading to TR [10]. Affected cats showed Ca or Mg deficiency [14].

Systemic disorders might be associated to TR in cats. The calcium—vitamin D—parathyroid hormone (PTH) system would play a role in the systemic activation of osteoclastic bone resorption; however, only serum concentration of 25-hydroxyvitamin D (25-OHD) was associated to TR, even though its role on etiopathogenesis of TR remains unclear [15,16]. On the other hand, cats with TR showed lower urine specific gravity but since a renal function test was not performed, an association of TR and impairment of renal function was not demonstrated [16].

The aim of the present study is twofold: (1) to describe the TR clinical presentation on the basis of individuals attending to odontology service at our University Veterinary hospital (Northeastern Spain), and (2) to study several blood parameters in TR cases and controls in an attempt to ascertain potential systemic effects associated with TR.

## 2. Materials and Methods

In this retrospective case control study, a 1:2 case control ratio group was used; although considering more than one control per case increases power, little increase is obtained when including more than two controls per case [17]. Cases and controls were matched for both geographical origin (Northeastern Spain) and period of presentation to hospital (September 2018–May 2019). Since increased age is an important predisposition factor for TR, it was not possible to match cases and controls for this criterion: controls must be necessarily free of any oral disease and, therefore, they were younger than cases. Therefore, age was considered as a correction factor in the applied regression models (see below). Figure 1 shows the flow diagram.

We certify that the best practice of veterinary care and legal and ethical requirements have been met regarding the humane treatment of animals described in the study. As a part of our admission protocol, informed consent for use of anonymous data is routinely obtained from owners, since the objectives of our University Veterinary hospital are assistance, education, and research. In accordance with Spanish legislation for animal protection in non-experimental veterinary procedure [18], approval by the local Ethical Committee was not needed for the present study, as stated by the Ethical Advisory Commission for Animal Experimentation of the University of Zaragoza (Ref. PI40/21NE).

The inclusion criteria for TR cases were: (1) positive diagnoses at the University Veterinary hospital of Zaragoza (Spain) between September 2018 and May 2019; a cat was considered as TR-affected if it showed at least one affected tooth and (2) biochemical and hematological data were available. These data were obtained on the basis of clinical decisions and we just recorded them. The presence of tumours and jaw fractures were exclusion criteria. The TR cases group included 29 cats.

For controls, the inclusion criterion was the availability of biochemical and haematological data at the time of hospital admission. Biochemical and haematological analytics is usually part of our pre-anesthetic protocol in cats. Exclusion criteria were the presence of any oral pathology, chronic diseases, or previous treatments with antibiotics, corticosteroids, and/or non-steroidal anti-inflammatory drugs. Cats included in the control group (58 individuals) were undergoing elective surgery (40/58; 69%), orthopedic surgery (11/58; 19%), or soft tissues procedures (7/58; 12%) in our university veterinary hospital.

Every cat was weighed and information about age, presence of halitosis, spontaneous bleeding (reddish saliva), and excessive drooling was obtained from the owner. Age was measured in years (y) and grouped in three categories [19]: young (0.2–6 y); mature (7–10 y), and senior/geriatric (11–16 y).

The cats were anesthetized to enable their dental examination. Premedication consisted of an intramuscular combination of dexmedetomidine (Dexdomitor, 0.5 mg/mL, Ecuphar Veterinaria, Barcelona, Spain), ketamine (Anesketin 100 mg/mL, Dechra Veterinary Solutions, Barcelona, Spain), and an opioid [methadone (Semfortan 10 mg/mL, Dechra Veterinary Solutions, Barcelona, Spain), buprenorphine (Buprenodale 0.3 mg/mL, Dechra Veterinary Solutions, Barcelona, Spain) or butorphanol (Torbugesic Vet 10 mg/mL, Zoetis Spain, Madrid, Spain)]. Propofol or Alfaxalone was used for intravenous induction until intubation was allowed and anesthetic maintenance with isoflurane was performed (IsoFlo, 100% p/p, Ecuphar Veterinaria, Barcelona, Spain). When indicated, locoregional anesthesia was administered (e.g., maxillary or mandibular blocks in the case of extraction of dental pieces or intra-testicular blocks in the case of orchiectomy).

TR was detected by means of a full-mouth radiography using a high-frequency X-ray tube (Toshiba, DG-073-B, 70kv, 8 mA) and an imaging plate scanner (Planmeca ProScanner). The stage (1–5) and type (1–3) of TR for each tooth were determined according to the American Veterinary Dental College (AVDC) recommendations [20]. Table 1 and Table 2 show the characteristics of each stage and type, respectively. Unaffected TR teeth were considered to be stage 0. Diagnoses of both periodontal disease (PD) and feline chronic gingivostomatitis (FCGS) were also according to the AVDC recommendations, as described elsewhere [20]. All the evaluations were performed by A.W., an experienced dentist. Data were recorded on individual dental charts. Nomina Anatomica Veterinaria (NAV) system of teeth naming was used (U: Upper, maxillary; L: Low, mandibular; R: Right side; L: Left side; I: Incisor; C: Canine: P: Premolar; M: Molar; 1, 2, 3…: number of tooth).

To assess the global state of each individual’s mouth, a TR global index was developed:

TR global index= (n_0_x0 + n_1_x1 + n_2_x2 + n_3_x3 + n_4_x4 + n_5_x5)/(n_0_ + n_1_ + n_2_ + n_3_ + n_4_ + n_5_)
where n_0_, n_1_, n_2_, n_3_, n_4_, and n_5_ mean the number of teeth for the stages 0, 1, 2, 3, 4, and 5, respectively, for each cat. Therefore, the TR global index is a continuous variable in ranges 0 (no tooth affected by TR) to 5 (all teeth in TR stage 5).

In our University Veterinary Hospital, biochemical and haematological analytics is usually part of the pre-anesthetic protocol in cats, especially in dentistry patients. This analytics included complete blood count (CBC) (Idexx LaserCyte) and serum chemical analysis (IDEXX Catalyst Dx); a total of 34 blood parameters were tested.

Statistical analysis was generated by the SPSS software v.22 (IBM). About the TR clinical presentation, chi squared test (χ^2^) was used to assess the relationship between TR stage and TR type on the basis of the tooth (H_0_: there is no relationship between TR stage and TR type). When a significant association was found, Cramer’s V determined the strength of this association. Percentage of males and age groups were compared between cases and controls by mean of χ^2^ (H_0_: there is no difference between cases and controls for these percentages).

The considered blood parameters were submitted to the Shapiro-Wilks test to assess their normal distribution. The median and interquartile range (IQR) were used to summarize them. To test the association of these blood parameters with TR, a two-phase analysis was performed:

(1) Non-parametric tests (U Mann-Whitney test, Kruskal-Wallis test) were carried out to compare age, weight, and the blood parameters measures between cases and controls. These variables were not normally distributed and therefore non-parametric tests were the most suitable methods to carry out such comparisons. Spearman’s rho measured the strength of association between each blood parameter and the TR global index. The global sample size (87 individuals) provided enough power to detect significant rho values of at least 0.3 (absolute value, power = 0.81; *p <* 0.05) [21]. According to Cohen [22], strength of association between quantitative variables was considered as small (0.1 < absolute r value <0.3), moderate (0.3 < absolute r value < 0.5), and large (absolute r value > 0.5). *p*-values < 0.050 were considered statistically significant.

(2) Once all available blood parameters were examined, those showing significant association with TR were chosen for multiple regression analysis. In this context, multiple regression analysis has been used for analyzing the association of TR (TR affected/TR unaffected; TR global index) with several independent variables (set of blood parameters and age as correcting factor). According to the TR assessment, two types of multiple regression analysis were performed.

A binomial logistic regression was performed to ascertain the association of age and blood parameters (independent variables) with affected TR individual/unaffected TR individual (dependent dichotomous variable). A cat was considered as an affected TR individual if showing at least one affected tooth. A stepwise procedure (Method: forward) was applied, where independent variables moved in or out of the model at any step of the process, on the basis of the Wald test, which determined statistical significance for each of the independent variables: the significance levels to enter and to be removed were *p* ≤ 0.05 and *p* ≥ 0.10, respectively.

Sample sizes for TR cases (29) and controls (58) provided a power of about 0.80 for detecting a minimum odds ratio of approximately 3.5 for risk factors; also, these sample sizes enabled detection of odds ratios of approximately 0.17 or lower (protective factors) with a power of about 0.80 [23]. As suggested by Chen and others [24], an odds ratio of approximately 3.5 would be considered as a medium size odds ratio.

A hierarchical multiple regression was performed to determine if the addition of the chosen blood parameters improved the prediction of the TR global index (quantitative dependent variable) over age alone, meaning a significant association of these blood parameters with TR global index (Method: enter; criteria for blood parameters to enter and be removed: *p* ≤ 0.05 and *p* ≥ 0.10, respectively).

## 3. Results

In the 29 TR individuals, excessive drooling, halitosis, and spontaneous bleeding were observed in 4 (13.8%), 11 (37.9%), and 1 cats (3.4%), respectively. PD was present in 13 cats (44.8%), FCGS in 2 (6.9%), and both PD and FCGS in 10 (34.5%); only 4 individuals (13.8%) did not show signs of either PD or FCGS.

For cases, Table 3 shows both TR stage and type as proportions on individual tooth, total tooth type, and total teeth. No type could be identified for teeth in TR stage 5 and these are shown in column NI (not identified).

TR was detected in 130/870 teeth (14.9%) but no incisors were found to be affected. The most frequently observed TR stage was stage 4 and this was in tooth LLP1, while the highest TR stage 5 occurrence was in tooth LRP1, followed by toothLLP1. TR types 1 and 2 were the most frequent ones with teeth 207 and 208 most frequently affected by TR type 1 and teeth 304 and 404 by TR type 2.

A significant association was found (χ^2^ = 53.977; df = 6; *p <* 0.001) between TR stage and TR type assessed on the same tooth (see Table 4). A large size effect was detected, indicating an intense association strength (Cramer’s V = 0.715; *p <* 0.001). Stages 1 and 2 (milder stages) were clearly associated with type 1. Teeth in more serious stages (3 and 4) showed any of the three types, with predominance of type 1 for stage 3 and of type 2 for stage 4.

The TR global index was 0 for every control individual. For TR cases, the TR global index (variable not normally distributed; *p <* 0.001) ranged from 0.06 to 1.83, a median of 0.43 and an IQR of 0.48.

Table 5 shows the distribution of breeds, percentage of males, distribution of neutered individuals, distribution of age groups and median and IQR for age (years), weight, complete blood count, and serum chemical analysis from TR affected individuals and controls. Table 5 also shows Spearman’s rho values for quantitative variables and TR global index. No breed was over represented in either of the groups (*p =* 0.142). Cases showed significantly greater values for percentage of males, neutered individuals (both males and females), mature and senior/geriatric cats, age (years), weight, MCV, NEU%, NEU/LYM, CREA, TP, and GLOB (*p <* 0.05). Conversely, controls showed significantly greater values for RBC, HGB, RDW, LYM%, BUN/CREA, ALB, ALB/GLOB, ALT, and ALKP (*p <* 0.05). Also, significant and positive Spearman’s rho values were found for TR global index and age (years), weight, MCV, NEU%, NEU/LYM, CREA, TP, and GLOB (*p <* 0.05); therefore, TR global index tend to increase when these variables increase. For TR global index and RBC, HGB, RDW, LYM%, BUN/CREA, ALB, ALB/GLOB, ALT and ALKP significant and negative rho values were found (*p <* 0.05); therefore, TR global index tend to decrease when these variables increase.

The strength of association with TR global index was small for MCV, RDW, ALB, and ALKP; and it was moderate for RBC, LYM%, HGB, weight, NEU%, NEU/LYM, BUN/CREA, ALT, and TP and large for CREA, ALB/GLOB, GLOB, and age.

The final model for binomial logistic regression is shown in Table 6. This model was statistically significant (*p <* 0.001). Sex and age were significantly associated with TR presence. Males were 7.388 times more likely to exhibit TR than females (*p =* 0.040). Mature cats (7–10 y) were 35.029 times more likely to TR than young cats (0.2–6 y); no significant difference was found for senior/geriatric cats compared to young ones (*p =* 0.818). Increasing CREA values (*p* = 0.034) and decreasing BUN/CREA and ALB/GLOB values were associated with TR presence (*p =* 0.029 and *p =* 0.03, respectively).

The final model for hierarchical multiple regression is shown in Table 7. The full model included age and GLOB (*p <* 0.001); increasing age and GLOB was associated with increasing TR index. This model was statistically significant (*p <* 0.001).

## 4. Discussion

An important limitation of the present work is the small number of studied individuals, which was attended to at the facilities of just one clinic. However, as mentioned in Material and Methods, this sample size provided adequate power for the proposed statistical analysis. Further, the University Veterinary hospital of Zaragoza is the referral center of the area (Aragon, Northeastern Spain); therefore, the study samples can be considered as representative of the local cat population and identical diagnostic and analytical procedures were applied to each individual.

As shown in Results, teeth LLP1, LRP1, and LLM1 were the most affected. Similar results were previously reported [3]. Heaton and others found that LLP1 and LRP1 are usually the first teeth affected by TR and therefore they were considered as TR sentinel teeth [25]. In 13/29 cases (44.8%), LLP1 or LRP1 were affected and both LLP1 and LRP1 were affected in 12/29 (41.4%). Neither LLP1 nor LRP1 were affected in only 3/29 cases (10.4%); two of these cases showed LLM1 as the only TR affected tooth and in the third case, only LRC1 was TR affected. In one case (1/29), both LRP1 and LRP1 were missing but there were 5 TR-affected teeth, including LLP2 and LRC1. Our results corroborated the role of LLP1 and LRP1 as TR sentinels, but maybe LLM1 could be added as such.

The five TR stages vary from mild dental tissue loss (stage 1) to crown loss with only root remnants remaining (stage 5) [12]. Stages 4 and 5 were the most frequently observed; both progressive TR character and high percentage of affected cats older than 7 years would explain these findings. The detected association between TR stage and type corroborate the results from Niemiec, who associated type 2 lesions with later stages [26].

In a previous work in cats, Reiter and others studied the associations between presence of TR lesions and several epidemiologic and laboratory variables [16]. The novelties of the present study with respect to Reiter’s are the inclusion of more blood parameters and the consideration of TR stages by means of the TR index.

An association of TR and males has been detected in this study. In the literature, data on the relationship between sex and TR are inconclusive: some studies reported TR being more frequent in males [27], in females [4,28], or unaffected by the sex of the cat [16,29]. No increased risk of TR was detected in neutered cats [30]. In this retrospective case control study, males were more frequently observed in the TR than in the control group (see Table 5) and this difference could explain the detected association. On the other hand, the proportion of neutered individuals (males and females) was significantly higher in the TR than in the control group: most of the individuals in the control group went to hospital to be neutered (elective surgery) and this fact could explain for the observed bias. According to our experience in the University Veterinary hospital of Zaragoza, neutering is usually the objective of a young cat first visit; actually many individuals in the control group came to be neutered at our Hospital (elective surgery, as said in Material and Methods). However, older cats, mostly neutered in their youth, came to our hospital due to other health problems. Therefore, at least in our cases and controls, the proportion of neutered individuals could be related to age; older animals tend to be neutered more often than younger ones.

Cats in our control group were significantly younger than cats in the TR group. It is proven that the prevalence of TR increases with age [3,16,28,29], and that could explain the fact that cats in the control group were younger. On the other hand, this difference in age would explain the observed differences in weight and blood parameters [31]. This difference in age between cases and controls is an important limitation of the present study and we used age correction by means of regression models, which offer a more adequate estimation of the association between TR and a set of blood markers than simply applying statistical test for individual markers. Significant effects of age on TR were detected for both logistic and hierarchical multiple regression; however, binomial logistic regression failed to find significant differences for senior/geriatric cats, compared to young ones. A low proportion of senior/geriatric cats in the TR group could cause a decrease in the power of the statistical test that would explain this result.

Controls showed significantly greater values for RBC, HGB, RDW, ALB, ALT, and ALKP, but these values were always into reference ranges. For LYM%, BUN/CREA, and ALB/GLOB, no reference ranges have been defined, but values for WBC, BUN, CREA, ALB, and GLOB were into reference ranges for controls. On the other hand, all controls went through a careful analysis of their state of health, so that they were totally healthy individuals, which justified their inclusion as controls.

ALB/GLOB and GLOB were present in logistic and hierarchical regression models, respectively, pointing to an association of increased GLOB levels with both TR occurrence in some of the teeth in an individual and increasing severity of the global state of an individual’s mouth. No differences in ALB/GLOB or GLOB between TR affected and not affected TR cats were found by Reiter and others [16].

Corporal response against several aggressions (infection, trauma, cancer) is inflammation [32]. As a consequence of inflammation, the release of several proteins to bloodstream occurs; if their concentrations increase or decrease by at least 25%, they can be used as systemic inflammatory markers [33]. There are many inflammatory markers (known as acute phase reactants); the most commonly used in clinical practice are C-reactive protein (CRP), erythrocyte sedimentation rate (ESR), and procalcitonin (PCT) [34,35]. Determination of both CRP and ESR was not part of our usual routine, but PCT was. PCT is a newer marker supposed to be useful in identifying or excluding bacterial infections and guiding antibacterial treatments [34,35]. However, no significant difference was found between the TR and C groups for PCT, having values in the reference range.

Serum proteins include ALB, α, β, and γ GLOB [36]. Some acute phase proteins (APPs) (ceruloplasmin, haptoglobin, α-2-macroglobulin) are α GLOB. APPS are increased in acute inflammation and both malignancy and nephritic syndrome [37]. Acute phase proteins, specifically serum amyloid A, alpha-1 acid glycoprotein, and haptoglobin have been suggested as indicators of acute inflammatory response in cats [38]; however their determination is not part of our usual routine, as described in Material and Methods, and they were not studied. Transferrin, a negative APP, and lipoproteins, complement and immunoglobulins, mainly IgA or IgM are β GLOB; inflammation, neoplasia and various metabolic conditions show increased β GLOB [37]. The γ GLOB fraction include immunoglobulins, particularly IgG, and the APP C-reactive protein; therefore, chronic antigenic stimulation is associated with increased γ GLOB [33]. In cats, IgG is a key biomarker of the systemic inflammatory response and might be specific for oral pathogens [39].

GLOB (serum Globulins, mainly IgG) and serum ALT (Alkaline phosphatase) are key biomarkers of the systemic response in cats with PD [39]. Globulin levels are directly related to the severity of PD [39]. There are different hypotheses and explanations for the role of GLOB and, in particular, IgG: one study suggests that they might be specific for oral pathogens in cats [39]; in humans affected by periodontitis, it seems to be related to the degree of periodontal bone loss [40]; potentially, the extent of tissue destruction and related inflammation induce the production of autoantibodies in aggressive PD in humans [41].

A decreased ALB/GLOB ratio is a classic pattern present in different feline pathologies with infectious-inflammatory or immune-mediated etiology, possibly due to persistent antigenic stimulation and compensatory reduction in albumin concentration [42]; ALB is a negative acute phase protein whose decrease in inflammation supposes a saving of amino acid for a more efficient production of AP [43]. In humans, the ALB/GLOB ratio is a marker that also decreases in PD, compared to the controls [44]. Proteinogram evaluation is a valuable test that identifies the different populations of immunoglobulins, which could clarify the pathophysiology of the infectious/inflammatory processes; however, their determination is not part of our usual routine, as described in Material and Methods, and they were not studied.

The association of TR with both biomarkers of infectious/inflammatory processes would suggest a potential systemic effect associated with TR but on the other hand they could also be explained by the concomitant presence of PD and TR. Humans affected by PD suffer repeated episodes of bacteremia, as a result of minor trauma to sites of periodontal inflammation, and may produce infection or inflammation and disease stimulated at distant organs [45]. Furthermore, some authors also suggest that endotoxemia and immune responses to bacteria and their products might lead to distant immune and toxic diseases as well [46]. PD shows high prevalence in cats [47]; therefore, PD-affected teeth are subjected to inflammation processes that would activate osteoclastic activity [7,8]. In this study, PD was present in 23/29 TR-affected cats (79.3%). The concomitant presence of both PD and TR would point to the inflammatory origin of TR in the considered cases.

According to most authors, type 1 TR is highly related to inflammation and PD, while type 2 TR is genuinely idiopathic [3,28]. Although TR is not considered an inflammatory disease, Arzi and others detected a similar number of mast cells in the gingiva of cats affected by TR, PD, and FCGS, compared to healthy cats [48]. They suggested that the mild inflammatory reaction observed and the high number of mast cells in the TR group could play a role in the pathogenesis of TR [48]. Type 1 TR was the most frequently observed type in this study (5.7% of total teeth in TR cases); therefore, this could be explained by the detected association of TR and inflammation biomarkers.

Reiter and others detected significantly higher BUN concentration in affected TR cats but this would be due to differences in age between affected and not-affected TR cats [16]. In addition, higher CREA concentration was detected in TR-affected cats but the difference was not significant [16]. Therefore, Reiter and others did not demonstrate a relationship between TR and impaired kidney function [16]. Renal parameters might be altered in older cats [31], hence implying the importance of applying an age correction. In this study, after correction to the age factor in the binary logistic regression, TR has also been associated with two blood parameters considered as biomarkers of kidney damage (CREA and CREA/UREA). Symmetric dimethylarginine (SDMA) is a more sensitive biomarker than CREA for declining glomerular filtration rate in dogs and cats [49], but these data were not available for this retrospective study. For the first time, CREA and CREA/UREA have been associated with TR. PD-affected cats were more likely to have been diagnosed with chronic kidney disease (CKD) than healthy cats (OR = 1.8) [50].

The association of these biomarkers with PD has been found in other species besides cats, such as humans and dogs [51,52]. In the study of Pavlica and others in dogs with diagnosed PD, glomerular and interstitial changes in kidneys were observed and related to immune complex-mediated damage as a consequence of PD [53]. The concomitant presence of PD and TR would again cause this association. However, the association of TR with both biomarkers of kidney damage would suggest the need to subject cats affected by TR to a test of renal efficiency.

## 5. Conclusions

The teeth most frequently affected by TR were 307, 407, and 309. Type 2 lesions were associated to later stages. Four blood parameters associated with TR have been detected: GLOB, ALB/GLOB (biomarkers of infectious/inflammatory processes), CREA, and BUN/CREA (biomarkers of kidney damage). The increase of GLOB was associated with an increasing TR index. The increase of CREA and the decrease of both BUN/CREA and ALB/GLOB values were associated with TR presence. The detected associations between blood parameters and TR cannot be interpreted as a cause-and-effect relationship. These associations may not have a significance in prevention or understanding of causality of the TR. However, they may be a call for attention.

When affected by TR, systemic implication related to infection/inflammation or even kidney damage could be present; therefore, special care in these aspects must be provided in feline clinics.

## Figures and Tables

**Figure 1 animals-11-02125-f001:**
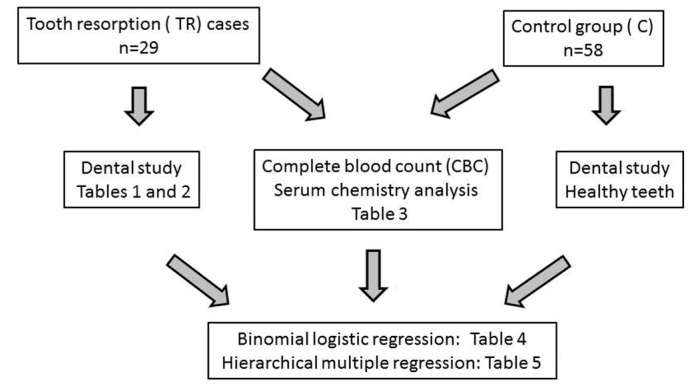
Flow diagram.

**Table 1 animals-11-02125-t001:** Staging (1–5) of TR (tooth resorption) according to American Veterinary Dental College (AVDC) recommendations [20].

Stage	Characteristics
TR1	Mild dental hard tissue lost (cementum or cementum and enamel)
TR2	Moderate dental hard tissue lost Loss of dentin not extended to pulp cavity Painful tooth if dentin tubules exposed to air
TR3	Deep dental hard tissue lost Loss of dentin extended to pulp cavity Integrity of the most of the tooth Very painful tool if exposed to air Bleeding from pulp tissue evident on probing Early “ghost images” for roots on radiographs
TR4	Extensive dental hard tissue lost Loss of dentin extended to pulp cavity Most of the tooth has lost its integrity TR4a: Crown and roots equally affected TR4b: More severe effects on the crown TR4c: More severe effects on the root
TR5	Remnants of hard dental tissues as irregular opacities Incomplete gingival covering Healed oral mucosa (sensitive or not) over the tooth fragments

**Table 2 animals-11-02125-t002:** Types (1–3) of TR (tooth resorption)according to American Veterinary Dental College (AVDC) recommendations [20].

Type	Characteristics
T1	Focal or multifocal radiolucency Normal radiopacity Normal periodontal ligament space
T2	Narrowing or disappearance of the periodontal ligament space Decreased radiopacity of part of the tooth
T3	Features of both type 1 and type 2 in the same tooth: Areas of normal and narrow or lost periodontal ligament space Focal or multifocal radiolucency Decreased radiopacity in other areas of the tooth

**Table 3 animals-11-02125-t003:** Description of TR (tooth resorption) stage and type in cases as proportions on individual tooth, total tooth type, and total teeth. NI (not identified: no type could be identified for teeth in TR stage 5); U (Upper, maxillary); L (Low, mandibular); R (Right side); L (Left side); I (Incisor); C (Canine); P (Premolar); M (Molar).

Tooth	TR Stage					TR Type			
	1	2	3	4	5	1	2	3	NI
Canines (Total tooth type)	5/116 (4.3%)	2/116 (1.7%)	1/116 (0.9%)	18/116 (15.5%)	1/116 (0.9%)	8/116(6.9%)	18/116 (15.5%)		1/116 (0.9%)
URC1				3/29 (10.3%)			3/29 (10.3%)		
ULC1		1/29 (3.4%)		3/29 (10.3%)		1/29 (3.4%)	3/29 (10.3%)		
LLC1	1/29 (3.4%)	1/29 (3.4%)	1/29 (3.4%)	6/29 (20.7%)	1/29 (3.4%)	3/29 (10.3%)	6/29 (20.7%)		1/29 (3.4%)
LRC1	4/29 (13.8%)			6/29 (20.7%)		4/29 (13.8%)	6/29 (20.7%)		
Premolars (Total tooth type)	7/290 (2.4%)	5/290 (1.7%)	25/290 (8.6%)	20/290 (6.9%)	20/290 (6.9%)	32/290 (11.0%)	16/290 (5.5%)	9/290 (3.1%)	20/290 (6.9%)
URP1			1/29 (3.4%)				1/29 (3.4%)		
URP2			4/29 (13.8%)	2/29 (6.9%)	1/29 (3.4%)	2/29 (6.9%)	3/29 (10.3%)	1/29 (3.4%)	1/29 (3.4%)
URP3			3/29 (10.3%)	2/29 (6.9%)	1/29 (3.4%)	2/29 (6.9%)	3/29 (10.3%)		1/29 (3.4%)
ULP1		1/29 (3.4%)				1/29 (3.4%)			
ULP2	2/29 (6.9%)	1/29 (3.4%)	3/29 (10.3%)			6/29 (20.6%)			
ULP3	2/29 (6.9%)	1/29 (3.4%)	3/29 (10.3%)	1/29 (3.4%)		6/29 (20.6%)		1/29 (3.4%)	
LLP1	1/29 (3.4%)		3/29 (10.3%)	8/29 (27.6%)	7/29 (24.1%)	5/29 (17.2%)	3/29 (10.3%)	4/29 (13.8%)	7/29 (24.1%)
LLP2	2/29 (6.9%)		1/29 (3.4%)	2/29 (6.9%)	2/29 (6.9%)	4/29 (13.8%)	1/29 (3.4%)		2/29 (6.9%)
LRP1		1/29 (3.4%)	5/29 (17.2%)	4/29 (13.8%)	8/29 (27.6%)	4/29 (13.8%)	4/29 (13.8%)	2/29 (6.9%)	8/29 (27.6%)
LRP1		1/29 (3.4%)	2/29 (6.9%)	1/29 (3.4%)	1/29 (3.4%)	2/29 (6.9%)	1/29 (3.4%)	1/29 (3.4%)	1/29 (3.4%)
Molars (Total tooth type)	3/116 (2.6%)	2/116 (1.7%)	4/116 (3.4%)	9/116 (7.7%)	8/116 (6.9%)	10/116 (8.6%)	2/116 (1.6%)	6/116 (5.2%)	8/116 (6.9%)
URM1									
ULM1	1/29 (3.4%)					1/29 (3.4%)			
LLM1	2/29 (6.9%)	2/29 (6.9%)	2/29 (6.9%)	6/29 (20.7%)	5/29 (17.2%)	7/29 (24.1%)	2/29 (6.9%)	3/29 (10.3%)	5/29 (17.2%)
LRM1			2/29 (6.9%)	3/29 (10.3%)	3/29 (10.3%)	2/29 (6.9%)		3/29 (10.3%)	3/29 (10.3%)
Total teeth	15/870 (1.7%)	9/870 (1.1%)	30/870 (3.4%)	47/870 (5.4%)	29/870 (3.3%)	50/870 (5.7%)	36/870 (4.2%)	15/870 (1.7%)	29/870 (3.3%)

**Table 4 animals-11-02125-t004:** Relationship between TR (tooth resoption) stage and TR type on the basis of the tooth. ^a,b^, Values within a row with different superscripts differ significantly at *p <* 0.050.

TR Type	TR Stage			
	1 (*n* = 15)	2 (*n* = 9)	3 (*n* = 30)	4 (*n* = 47)
1	15/15 (100%)^a^	9/9 (100%) ^a^	20/30 (66.7%) ^a^	6/47 (12.8%) ^b^
2	-	-	6/30 (20%) ^a^	30/47 (63.8%) ^b^
3	-	-	4/30 (13.3%) ^a^	11/47 (23.4%) ^a^

**Table 5 animals-11-02125-t005:** Characteristics (breed, sex, age, and weight), complete blood count and serum chemical analysis (with reference range) from TR (TR affected individuals, Cases) and C (Control) groups. Data are counts/n (%) for categorical variables, median [IQR: interquartile range] for quantitative not normally distributed variables, *p*-values for comparisons between TR and C groups (χ^2^, U Mann-Witney test, Kruskall-Wallis test) and Spearman’s rho for each variable and TR (tooth resorption)global index (*p*-value for rho), y: years. Values within a row with different superscripts differ significantly at *p <* 0.050.

Variable	Reference Range	TR (*n* = 29)	C (*n* = 58)	*p*-Value (Comparisons between TR and C groups)	Spearman’s Rho Variable and TR Global Index (*p*-Value)
Breed				0.142	
Domestic short hair		20/29 (69%)	51/58 (87.9%)		
Sphynx		1/29 (3.4%)	0/58 (0%)		
Maine Coon		1/29 (3.4%)	0/58 (0%)		
Persian		2/29 (6.9%)	4/58 (6.9%)		
Ragdoll		1/29 (3.4%)	0/58 (0%)		
Shorthair		1/29 (3.4%)	0/58 (0%)		
Siamese		3/29 (10.3%)	3/58 (5.2%)		
Males		20/29 (69%)	24/58 (41.4%)	0.015	
Neutered individuals					
Males		17/20 (85%)	6/24 (25%)	<0.001	
Females		9/9 (100%)	4/34 (11.8%)	<0.001	
Age group				<0.001	
Young (0.2–6 y)		6/29 (20.7%) ^a^	50/59 (86.2%) ^b^		
Mature (7–10 y)		12/29 (41.4%) ^a^	4/58 (6.9%) ^b^		
Senior/geriatric (11–16 y)		11/29 (37.9%) ^a^	4/58 (6.9%) ^b^		
Age (years)		9.00 [4.5]	1.90 [3.0]	<0.001	0.651 (<0.001)
Weight (kg)		4.35 [1.35]	3.40 [1.30]	0.001	0.333 (0.002)
Red blood cells (RBC)	6.54–12.20 M/µL	7.18 [2.07]	8.39 [2.13]	0.004	−0.309 (0.004)
Haematocrit (HCT)	30.3–52.3%	30.60 [6.35]	34.95 [9.95]	0.072	−0.206 (0.055)
Haemoglobin (HGB)	9.8–16.2 g/dL	11.00 [1.70]	12.30 [2.90]	0.002	−0.329 (0.002)
Mean corpuscular volume (MCV)	35.9–53.1fL	43.50 [8.0]	41.60 [5.8]	0.030	0.225 (0.036)
Mean corpuscular haemoglobin (MCH)	11.8–17.3 pg	14.73 [2.0]	14.55 [1.38]	0.249	0.124 (0.251)
Mean corpuscular haemoglobin concentration (MCHC)	28.1–35.8 g/dL	34.10 [2.7]	35.10 [1.9]	0.072	−0.190 (0.085)
Red cell distribution width (RDW)	15–27%	22.00 [3.75]	24.20 [4.28]	0.020	−0.245 (0.022)
Reticulocyte percentage (RETIC)		0.10 [0.10]	0.20 [0.20]	0.331	−0.117 (0.285)
White blood cells (WBC)	2.87–17.02 K/µL	9.22 [7.36]	8.03 [4.35]	0.615	0.099 (0.363)
Neutrophils percent (NEU%)		70.55 [20.3]	56.20 [23.7]	0.006	0.336 (0.002)
Lymphocytes percent (LYM%)		17.80 [17.85]	29.80 [19.00]	0.007	−0.321 (0.003)
Monocytes percent (MON%)		4.40 [2.50]	4.20 [2.80]	0.945	−0.031 (0.775)
Eosinophils percent (EOS%)		4.80 [5.90]	6.40 [5.60]	0.411	−0.124 (0.254)
Basophils percent (BASO%)		0.40 [0.53]	0.60 [0.50]	0.081	−0.204 (0.064)
Neutrophil to lymphocyte ratio (NEU/LYM)		3.88 [4.27]	1.85 [2.31]	0.003	0.352 (0.001)
Platelets (PLT)	151–600 K/µL	278 [251.5]	270 [235.8]	0.627	0.030 (0.784)
Platelets to lymphocyte ratio (PLT/LYM)		188.04 [121.02]	120.12 [157.02]	0.146	0.144 (0.187)
Mean platelet volume (MPV)	11.4–21.6 fL	16.25 [2.35]	16.20 [2.65]	0.808	−0.076 (0.491)
Mean platelet volume to platelets (MPV/PLT)		0.05 [0.05]	0.06 [0.05]	0.578	−0.056 (0.597)
Procalcitonin (PCT)	0–0.79 ng/mL	0.57 [0.41]	0.48 [0.36]	0.546	0.047 (0.670)
Glucose (GLU)	71–159 mg/dL	144.00 [75.0]	142.00 [77.0]	0.674	0.069 (0.526)
Blood urea nitrogen (BUN)	16–36 mg/dL	21.00 [10.0]	20.00 [5.5]	0.115	0.171 (0.112)
Creatinine (CREA)	0.8–2.4 mg/dL	1.70 [0.6]	1.20 [0.5]	<0.001	0.506 (<0.001)
BUN/CREA		13.00 [3.0]	16.00 [7.5]	0.002	−0.365 (0.001)
Total protein (TP)	5.7–8.9 g/dL	7.40 [1.4]	6.90 [1.3]	<0.001	0.426 (<0.001)
Albumin (ALB)	2.3–3.9 g/dL	2.70 [0.4]	3.00 [0.5]	0.015	−0.246 (0.022)
Globulin (GLOB)	2.8–5.1 g/dL	4.70 [1.3]	3.90 [0.7]	<0.001	0.547 (<0.001)
ALB/GLOB		0.60 [0.20]	0.80 [0.20]	<0.001	−0.536 (<0.001)
Alanine aminotransferase (ALT)	12–130 U/L	52.0 [35.0]	88.0 [67.0]	<0.001	−0.405 (<0.001)
Alkaline phosphatase (ALKP)	14–111 U/L	25.00 [27]	34.50 [50]	0.024	−0.251 (0.021)
Na	150–165 mmol/L	154.00 [4.8]	155.00 [6.0]	0.203	−0.114 (0.295)
K	3.5–5.8 mmol/L	3.85 [0.6]	3.80 [0.5]	0.993	0.044 (0.688)
Na/K		41.00 [5.0]	40.00 [4.3]	0.857	−0.013 (0.907)
Cl	112–129 mmol/L	116.00 [7.0]	117.00 [6.0]	0.397	−0.018 (0.869)

**Table 6 animals-11-02125-t006:** Final model for binary logistic regression, showing the significant associations of TR (tooth resorption) to sex, age, and serum variables CREA, BUN/CREA, and ALB/GLOB. B: regression coefficient; SE: standard error; df: degrees of freedom; 95%CI: 95% interval of confidence.

	B	SE	Wald	df	*p*-Value	Odds Ratio	95% CI for Odds Ratio	
							Lower	Upper
Sex (Male)	2.000	0.973	4.221	1	0.040	7.388	1.097	49.782
Age			8.669	2	0.013			
Mature	3.556	1.208	8.666	1	0.003	35.029	3.282	373.828
Seniors/geriatric	0.263	1.140	0.053	1	0.818	1.300	0.139	12.153
CREA	2.614	1.235	4.480	1	0.034	13.658	1.213	153.729
BUN/CREA	−0.412	0.189	4.758	1	0.029	0.663	0.458	0.959
ALB/GLOB	−9.121	3.079	8.777	1	0.003	<0.001	<0.001	0.046
Constant	5.572	3.126	3.177	1	0.075	262.884		

**Table 7 animals-11-02125-t007:** Final model for hierarchical regression, showing the significant associations of TR (tooth resorption) global index to age, and serum variable Globulins. B: regression coefficient; SE: standard error; 95%CI: 95% interval of confidence for B.

Variable	B	SE	T	*p*-Value	95% CI for B	
					Lower	Upper
Constant	−0.732	0.159	−4.612	<0.001	−1.048	−0.416
Age	0.025	0.008	3.170	0.002	0.009	0.040
Globulins	0.194	0.041	4.688	<0.001	0.112	0.276

## Data Availability

Data supporting the reported results can be sent to anyone interested by contacting the corresponding author.

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
