# Peer review of "Blood Parameters and Feline Tooth Resorption: A Retrospective Case Control Study from a Spanish University Hospital"

_animals, 2021, doi:10.3390/ani11072125_

Round 1
Reviewer 1 Report
The manuscript by Ana Whyte et al., summarizes the stages and types of tooth resorptions (TR) and investigates potential correlations between TR and blood parameters on a feline population from Veterinary Teaching Hospital in Northeastern Spain. The authors report that TR were associated with elevated CREA and decreased BUN/CREA and ALB/GLOB.
The authors discuss the potential contribution of the periodontal and kidney disease to this association.
Overall, the manuscript is informative and has relevance to the field in that it provides a documentation of TR in Spanish population of cats. The impact of the study is average because detection of association may not have a significance in prevention or understanding of causality of the TR. Nevertheless, with minor corrections this manuscript is useful to the field of veterinary dentistry.
I suggest the following corrections to be performed:
- Throughout the manuscript and tables, please correct the Triadan tooth numbering system to the NAV system of teeth naming (i.e., instead of numbers use R/L incisor, premolar, molar, 1, 2, 3 tooth)
- Throughout the discussion, please correct the sentence segments to full sentences (in lines 331-334; line 368)
- In line 238-239 the authors report significantly increased values of ALT, ALKP, RBC, and other blood parameters in control cats. Please explain why these parameters were elevated in control population and why these cats were permitted to serve as controls in this study.
- For completeness of this work please provide an additional table summarizing the criteria for types of TR. Similar to what was presented in table 1. With both tables in place this MS can serve as a convenient reference for TR classification.
- Line 135: remove the words “more important”
- Lines 299-302: Either remove or further develop the paragraph. Since authors did not evaluate calciotropic hormones, comparison to work by Reiter et. al., is not relevant.
- Line 312: Please remove or provide reference to the statement that older individuals are neutered more often than younger ones.
- Line 369: replace the word submitted by subjected.
- Line 374: replace the word isolated by detected.
Author Response
Answer to reviewer 1:
The impact of the study is average because detection of association may not have a significance in prevention or understanding of causality of the TR.
Now it is said in Conclusions:
The detected associations between blood parameters and TR cannot be interpreted as a cause-and- effect relationship. These association may not have a significance in prevention or understanding of causality of the TR. However, they may be a call for attention.
- Throughout the manuscript and tables, please correct the Triadan tooth numbering system to the NAV system of teeth naming (i.e., instead of numbers use R/L incisor, premolar, molar, 1, 2, 3 tooth)
Done.
Furtermore, it is said now in Material and Methods: Nomina Anatomica Veterinaria (NAV) system of teeth naming was used (U: Upper, maxillary; L: Low, mandibular; R: Right side; L: Left side; I: Incisor; C: Canine: P: Premolar; M: Molar; 1,2,3…: number of tooth).
Table 3 shows these corrected teeth names; for clarity, these new names appear on yellow background. Also, in this table caption it is said now: ); U (Upper, maxillary); L (Low, mandibular); R (Right side); L (Left side); I (Incisor); C (Canine); P (Premolar); M (Molar).
- Throughout the discussion, please correct the sentence segments to full sentences (in lines 331-334; line 368
Now it is said :
- Serum proteins include ALB, a, b and g GLOB [36]. Some acute phase proteins (APPs) (ceruloplasmin, haptoglobin, a-2-macroglobulin) are a GLOB. APPS are increased in acute inflammation and both malignancy and nephritic syndrome [37].
-PD shows high prevalence in cats [47]; therefore PD affected teeth are subjected to inflammation processes that would activate osteoclastic activity [7, 8]
- In line 238-239 the authors report significantly increased values of ALT, ALKP, RBC, and other blood parameters in control cats. Please explain why these parameters were elevated in control population and why these cats were permitted to serve as controls in this study
Now it is said in Discussion: Controls showed significantly greater values for RBC, HGB, RDW, ALB, ALT and ALKP, but these values were always into reference ranges. For LYM %, BUN/CREA and ALB/GLOB, no reference ranges have been defined, but values for WBC, BUN, CREA, ALB and GLOB were into reference ranges for controls. On the other hand, all controls went through a careful analysis of their state of health, so that they were totally healthy individuals, which justified their inclusion as controls.
Note that the rest of discussion focuses on the increase of CREA and the decrease of both BUN/CREA and ALB/GLOB, related to TR, were the main conclusions of this work.
- For completeness of this work please provide an additional table summarizing the criteria for types of TR. Similar to what was presented in table 1. With both tables in place this MS can serve as a convenient reference for TR classification.
Done. Now, it is said in Material and methods: Tables 1 and 2 show the characteristics of each stage and type, respectively.
Hence, tables have been renumbered.
- Line 135: remove the words “more important”
Done.
- Lines 299-302: Either remove or further develop the paragraph. Since authors did not evaluate calciotropic hormones, comparison to work by Reiter et. al., is not relevant.
Now it is said :
In a previous work in cats, Reiter and others studied the associations between presence of TR lesions and several epidemiologic and laboratory variables [16]. The novelties of the present study with respect to Reiter´s are the inclusion of more blood parameters and the consideration of TR stages by means of TR index
- Line 312: Please remove or provide reference to the statement that older individuals are neutered more often than younger ones.
Now it is said :
According to our experience in the University Veterinary hospital of Zaragoza, neutering is usually the objective of a young cat first visit; actually many individuals in the control group came to be neutered at our Hospital (elective surgery, as said in Material and methods). However, older cats, mostly neutered in their youth, came at our Hospital due to other health problems. Therefore, at least in our cases and controls, the proportion of neutered individuals could be related to age; older animals tend to be neutered more often than younger ones.
- Line 369: replace the word submitted by subjected.
Done
- Line 374: replace the word isolated by detected.
Done
Reviewer 2 Report
This manuscript describes that the blood parameters and feline tooth resorption.
Generally, this manuscript is interesting. However, there are some concerns as presented and some of these are discussed below.
Specific points:
- The table is hard to see, so I think it's better to correct it.
- What do you think about other blood parameters?
- To what extent do you think local inflammation affects blood parameters?
- Overall, this is an interesting article and a novel field for further research.
- In its present form, the manuscript does not constitute a comprehensive article. In the current form I cannot support publication, but improvements can easily be implemented and should be considered as suggested above.
Author Response
Answer to reviewer 2:
Generally, this manuscript is interesting. However, there are some concerns as presented and some of these are discussed below.
Specific points:
- The table is hard to see, so I think it's better to correct it.
In according to reviewer 1 suggestions, a new table has been added and therefore, tables have been renumbered. Indeed, tables 3 and 5 are very big ones, but showing all information about teeth and both individual characteristics and blood analysis data is needed for best comprehension of our work. There is no leeway to reduce or simplify these tables. However, we hope that the final version of this article, if it would be finally accepted for publication, be edited by those responsible for this, so that these tables appear on a single page each, with adequate format. In this way, they would become easier to see.
On the other hand, tables 6 and 7 are written as usual in statistical works.
- What do you think about other blood parameters?
As said in Material and Methods, in our University Veterinary Hospital, biochemical and haematological analytics is usually part of the pre-anesthetic protocol in cats, especially in dentistry patients. This analytics included complete blood count (CBC) (Idexx LaserCyte) and serum chemical analysis (IDEXX Catalyst Dx); a total of 34 blood parameters were tested. These parameters are showed in the big table 5,
Also, as said in Material and Methods, inclusion criterium for both cases and controls was the availability of these biochemical and haematological data. No other blood parameters were available for us.
In this respect, now it is said in Discussion: There are many inflammatory markers (known as acute phase reactants); the most commonly used in clinical practice are C-reactive protein (CRP), erythrocyte sedimentation rate (ESR), and procalcitonin (PCT) [34, 35]. Determination of both CRP and ESR was not part of our usual routine, but PCT was. PCT is a newer marker supposed to be useful in identifying or excluding bacterial infections and guiding antibacterial treatments [34, 35]. However, no significant difference was found between TR and C groups for PCT, being values into reference range.
Also, it is said in discussion now: Acute phase proteins, specifically serum amyloid A, alpha-1 acid glycoprotein and haptoglobin have been suggested as indicators of acute inflammatory response in cats [38]; however their determination is not part of our usual routine, as described in Material and Methods, and they were not studied.
- To what extent do you think local inflammation affects blood parameters?
Now it is said in Discussion: Corporal response against several aggressions (infection, trauma, cancer) is inflammation [32]. As a consequence of inflammation, release of several proteins to bloodstream occurs; if their concentrations increase or decrease by at least 25%, they can be used as systemic inflammatory markers [33]. There are many inflammatory markers (known as acute phase reactants); the most commonly used in clinical practice are C-reactive protein (CRP), erythrocyte sedimentation rate (ESR), and procalcitonin (PCT) [34, 35]. Determination of both CRP and ESR was not part of our usual routine, but PCT was. PCT is a newer marker supposed to be useful in identifying or excluding bacterial infections and guiding antibacterial treatments [34, 35]. However, no significant difference was found between TR and C groups for PCT, being values into reference range.
Serum proteins include ALB, a, b and g GLOB [36]. Some acute phase proteins (APPs) (ceruloplasmin, haptoglobin, a-2-macroglobulin) are a GLOB. APPS are increased in acute inflammation and both malignancy and nephritic syndrome [37]. Acute phase proteins……
As a consequence four new references have been added and reference list has been renumbered:
[32]Ansar, W.; Ghosh, S. Inflammation and inflammatory diseases, markers, and mediators: Role of CRP in some inflammatory diseases. In: Biology of C reactive protein in health and disease; Springer; New Delhi, India, 2016; pp67-107. https://doi.org/10.1007/978-81-322-2680-2_4
[33] Gabay, C; Kushner, I. Acute-phase proteins and other systemic responses to inflammation. N Engl J Med 1999, 340, 448–454. doi: 10.1056/NEJM199902113400607.
[34] Meisner, M. Update on procalcitonin measurements. Ann Lab Med 2014, 34, 263-273. https://doi.org/10.3343/alm.2014.34.4.263
[35] Schuetz, P.; Beishuizen, A.; Broyles, M.; Ferrer, R.; Gavazzi, G.; Gluck, E.H.; González Del Castillo, J.; Jensen, J.U.; Kanizsai, P.L.; Kwa, A.L.H.; Krueger, S.; Luyt, C.E.; Oppert, M.; Plebani, M.; Shlyapnikov, S.A.; Toccafondi, G.; Townsend, J.; Welte, T.; Saeed, K. Procalcitonin (PCT)-guided antibiotic stewardship: an international experts consensus on optimized clinical use. Clin Chem Lab Med 2019, 57(9):, 1308-1318. doi: 10.1515/cclm-2018-1181. PMID: 30721141.
- Overall, this is an interesting article and a novel field for further research.
- In its present form, the manuscript does not constitute a comprehensive article. In the current form I cannot support publication, but improvements can easily be implemented and should be considered as suggested above.
This reviewer is no very concrete in the suggestions; therefore we have done our best to interpret these suggestions. We hope that we have complied with these suggestions; otherwise, we would greatly appreciate receiving more detailed instructions
Round 2
Reviewer 2 Report
I am now satisfied that all necessary changes have been made.